

# Precise water level measurements using low-cost GNSS antenna arrays

David J. Purnell[1], Natalya Gomez[1], William Minarik[1], David Porter[2], and Gregory Langston[1]

[1]Department of Earth and Planetary Sciences, McGill University, 3450 University Street, Montreal, Quebec, Canada, H3A 0E8

[2]Lamont-Doherty Earth Observatory, Columbia University, New York, NY, USA

**Correspondence:** David Purnell (david.purnell@mail.mcgill.ca)

**Abstract.** We have developed a ground-based Global Navigation Satellite System Reflectometry (GNSS-R) technique for monitoring water levels with a comparable precision to standard tide gauges (e.g., pressure transducers) but at a fraction of the cost and using commercial products that are straightforward to assemble. As opposed to using geodetic-standard antennas that have been used in previous GNSS-R literature, we use multiple co-located low-cost antennas to retrieve water levels via inverse modelling of Signal-to-Noise ratio data. The low-cost antennas are advantageous over geodetic-standard antennas because they are much less expensive (even when using multiple antennas in the same location) and they can be used for GNSS-R analysis over a greater range of satellite elevation angles. We validate our technique using arrays of four antennas at three test sites with variable tidal forcing and co-located operational tide gauges. The root mean square error between the GNSS-R and tide gauge measurements ranges from 0.7–1.2 cm when using all four antennas at each site. We find that using four antennas instead of a single antenna improves the precision by 30–50% and preliminary analysis suggests that four appears to be the optimum number of co-located antennas. In order to obtain precise measurements, we find that it is important for the antennas to track GPS, GLONASS and Galileo satellites over a wide range of azimuth angles (at least 140 degrees) and elevation angles (at least 30 degrees).

## 1 Introduction

Precise water level measurements are needed for monitoring the global oceans, lakes and rivers, all of which are vulnerable to anthropogenic climate change (Goudie, 2006; Adrian et al., 2009; Slangen et al., 2016). Sea level at any location is influenced by spatiotemporally variable processes such as tides, storm surges and glacial isostatic adjustment (Tamisiea et al., 2014; Kopp et al., 2015). Networks of coastal water level sensors are therefore necessary to validate models of these processes (Meyssignac et al., 2017; Seifi et al., 2019; Dullaart et al., 2020) and satellite altimetry measurements (Gómez-Enri et al., 2018; Peng and Deng, 2020). The need for precise water level measurements was recently highlighted by Taherkhani et al. (2020), who showed that even a modest sea level rise of 1 cm in some regions could double the odds of a 50-year extreme water level event (i.e.,





a flooding event). The risk posed by extreme water level events is especially concerning given that current projections of globally averaged sea level rise by 2100 due to mass changes from the Antarctic ice sheet alone vary from 0 to 1.7 m (Pattyn

and Morlighem, 2020), whilst projections of mass loss from Greenland's largest outlet glaciers for the same period may be underestimated (Khan et al., 2020). Continuous records of sea level changes in the polar regions, that could be used to constrain the response of ice sheets to ongoing climate change, remain critically sparse (Baumann et al., 2020).

A range of different instruments are commonly used for monitoring water levels with variable cost and accuracy (GLOSS, 2012; Míguez et al., 2005; Pytharouli et al., 2018). With a budget of approximately 1,000–10,000 USD, it is possible to buy

an acoustic gauge or a pressure gauge to monitor water levels with sub-cm accuracy - the level of accuracy required for the Global Sea Level Observing System (GLOSS) network (GLOSS, 2012). However, pressure gauges may suffer from drift over multi-year timescales (Míguez et al., 2005; Pytharouli et al., 2018) and acoustic gauges are difficult to install in remote regions because they require a structure to hang over the water surface. Global Navigation Satellite System Reflectometry (GNSS-R) is an alternative technique to monitor water levels using geodetic-standard antennas that were designed to monitor land

deformation. These instruments can be purchased within the same budget as previously mentioned for acoustic or pressure sensors and do not suffer from the same issues. There are already many geodetic-standard antennas installed in remote regions to monitor earth deformation and a recent study demonstrated that coastal antennas in Greenland and Antarctica could also be used to monitor sea level Tabibi et al. (2020). However, in previous studies the precision of GNSS-R water level measurements was found to be greater than 1 cm and the datum of measurements is generally undefined (Larson et al., 2013; Strandberg et al.,

2016; Tabibi et al., 2020; Purnell et al., 2020).

Recently, as part of a broader trend in environmental sensing, there has been interest in the use of mobile devices and low-cost instrumentation for monitoring water levels. For example, Sermet et al. (2020) used images captured on smartphones to make river stage measurements and Strandberg and Haas (2019) applied GNSS-R techniques to make sea level measurements using the built-in GNSS antenna on a tablet computer. The latter study found comparable precision to GNSS-R measurements

from a co-located geodetic-standard antenna. Furthermore, Fagundes et al. (2020) applied GNSS-R techniques to monitor a lake in Brazil using instruments that cost approximately 200 USD and found a Root Mean Square Error (RMSE) of 2.9 cm when comparing with measurements from a co-located radar gauge.

Low-cost GNSS antennas such as those used by Strandberg and Haas (2019) and Fagundes et al. (2020) have shown to be better suited for GNSS-R than geodetic-standard antennas because geodetic-standard antennas are designed to reduce multipath

interference (the signal that is analyzed for GNSS-R measurements). Geodetic-standard antennas can only be used to make water level measurements at low elevation angles (often less than 20 degrees), whereas low-cost antennas are designed for mobile devices hence they are approximately isotropic in their gain pattern and can be used at larger elevation angles. The extra data at larger elevation angles is useful because the bias caused by tropospheric delay (or atmospheric refraction) is reduced at larger elevation angles (Santamaría-Gómez and Watson, 2017; Williams and Nievinski, 2017; Nikolaidou et al.,

2019). According to Nikolaidou et al. (2019), the tropsospheric delay bias tends to 0 for an antenna that is 1 – 10 m above a reflecting surface when using elevation angles larger than 20 degrees. Using data collected at elevation angles larger than 20 degrees could therefore eliminate the need for complicated tropospheric delay corrections that rely on global models with poor





spatial resolution (Williams and Nievinski, 2017) or additional instrumentation to make in-situ measurements (Santamaría-Gómez and Watson, 2017).

Purnell et al. (2020) recently showed that random noise in the Signal-to-Noise Ratio (SNR) is one of the dominant sources of uncertainty in GNSS-R water level measurements, particularly at high elevation angles. Their results suggest that multiple co-located antennas could be used to cancel out the effect of random noise in the SNR data and improve the precision of water level measurements. It would be prohibitively expensive to co-locate several geodetic-standard antennas for the purpose of cancelling out the effect of random noise, but multi-frequency, weatherproof GNSS antennas are commercially available

online for 10-30 USD, meaning that multiple low-cost antennas are still a fraction of the cost of a single geodetic-standard antenna.

  The purpose of this study is to test the hypothesis that multiple co-located antennas can be used to improve the precision of GNSS-R water level measurements and to demonstrate the effectiveness of low-cost antennas. We test this hypothesis by retrieving water level measurements from arrays of four stacked low-cost antennas at three different locations with variable tidal

forcing and compare the measurements with nearby operational tide gauges. Section 2 contains a summary of the technique that was developed by Strandberg et al. (2016) to retrieve sea level measurements using inverse modelling of SNR data and a description of how we adapted it for using multiple co-located antennas. In Section 3 we provide a description of the arrays of four stacked low-cost antennas that we used to retrieve water level measurements and in Section 4 we describe the three test sites. Finally in Section 5 we present our results from the test sites and in Section 6 we discuss a range of parameters related to

the GNSS-R analysis in order to guide future installations.

## 2 Inverse modelling of SNR data using multiple antennas

An antenna with a view of a water surface simultaneously receives GNSS signals that travel directly from a satellite and signals that reflect off the water surface prior to reaching the antenna. As a satellite moves in orbit, the difference in path length between the direct and reflected signals changes and the signals arrive at the antenna periodically in and out of phase, thereby

causing an oscillation in the SNR. For each period that a GNSS satellite is aligned with the water surface such that the antenna receives reflected signals, Larson et al. (2013) showed that the SNR can be represented as a function of the elevation angle of the satellite and the height of the antenna above the reflecting surface;

$$\delta\text{SNR} = A\sin\left(\frac{4\pi h}{\lambda}sin\theta + \phi\right), \tag{1}$$

where $\delta$SNR is the detrended SNR, $A$ depends on the power of the reflected signal and the antenna gain pattern, $h$ is the

reflector height, $\lambda$ is the wavelength of the GNSS signal, $\theta$ is the satellite elevation angle and $\phi$ depends on properties of the reflecting surface and the antenna phase response. The reflector height refers to the vertical distance between the antenna phase center and the reflecting surface, hence this value increases as the water level decreases. The step of detrending the SNR refers to removing a second order polynomial in $\sin\theta$ space. This step removes the influence of the antenna gain pattern and the position of the satellite.



Given the relationship in Equation 1, Strandberg et al. (2016) showed that the water level (analogous to $h$) can be retrieved via inverse modelling of SNR data. First, Equation 1 is modified for numerical stability by representing the oscillation as a summation of a sine and cosine wave,

$$\delta\text{SNR} = \left( C_1 \sin\left( \frac{4\pi h}{\lambda} \sin\theta \right) + C_2 \cos\left( \frac{4\pi h}{\lambda} \sin\theta \right) \right) e^{-4k^2 s^2 \sin^2\theta} \tag{2}$$

where $k = 2\pi/\lambda$ is the wavenumber of the GNSS signal, $s$ is related to the standard deviation of the reflecting surface height

and both $C_1$ and $C_2$ are related to $A$ and $\phi$. The damping factor at the end of equation 2 follows from Beckmann and Spizzichino (1987) and accounts for the loss of coherence of the reflected signal from a rough surface with standard deviation of $s$. For a predetermined period of consecutive data, referred to henceforth as a time window, the parameters $C_1$, $C_2$ and $s$ are assumed to be constant and $h$ is represented by a b-spline curve with a pre-determined knot spacing (for more information on the b-spline formulation, refer to Strandberg et al. (2016)). The parameters $C_1$, $C_2$ and $s$ and b-spline scaling factors (or node values)

are estimated simultaneously by reducing the residual between observed and modelled $\delta$SNR using a least squares algorithm. The parameters $C_1$, $C_2$ and $s$ are to be estimated once for each GNSS signal and satellite constellation used. The analysis is repeated and parameters are re-retrieved for each consecutive time window over the period of interest. As per Strandberg et al. (2016), only the scaling factors within the middle period of each time window (e.g., the middle day for a time window of 3 days) are used to form the final sea level time series in order to avoid instabilities at the ends of the b-spline curves.

As an additional step, we have also found it improves results to normalize the SNR data. This is done by scaling each period of detrended SNR data for each satellite prior to the inverse modelling analysis such that the absolute maximum value is always 100 (the number 100 is chosen arbitrarily). This is because the amplitude of the interference in the SNR data varies greatly between different satellite constellations; it is generally stronger for GLONASS satellites. Therefore, if the SNR data is not normalized, the results will be biased towards matching the data from GLONASS satellites (any residual between observed

and modeled SNR will be larger for GLONASS and hence will be prioritized over data from other satellites).

The formulation described above is adapted to account for multiple GNSS antennas in the same location as follows. For multiple co-located antennas at fixed heights relative to each other, changes in the geometric (real) reflector height for each antenna are equal but they are offset by a constant value. These constant offsets can be estimated by measuring the distance between each antenna, but this distance does not take into account possible variations in the antenna phase center, i.e., the

datum of the reflector height measurements. Instead, we retrieve a reflector height time series for each antenna and calculate the mean separation between each antenna. We then arbitrarily assign one antenna to be the reference antenna and remove the mean separation from the rest of the antennas to the reference antenna. With the adjusted reflector height solutions, we take a median of each b-spline scaling factor to produce the final, combined reflector height time series.

The effectiveness of the inverse modelling approach is dependant on several factors identified by Strandberg et al. (2016).

Firstly the initial choice of values for the parameters to be estimated in the least squares algorithm is important to find the global minimum solution as opposed to a local minimum. In this regard, the initial estimates of scaling factors should be informed by performing spectral analysis on the SNR data. An initial estimate of the reflector height is obtained using the equation

$$f = \frac{2h}{\lambda}, \tag{3}$$





Where f is the frequency of oscillations and $\lambda$ is the GNSS carrier wavelength. The median value of $h$ can be used as an initial estimate for all the scaling factors if there are negligible tides at a location ($< 0.1$ m), as is the case for one of our test sites. For sites a greater daily tidal range, the scaling factor estimates should be obtained by reducing the residual between the left and right hand side of the following equation from Larson et al. (2013)

$$\frac{\lambda f}{2} = h + \frac{\partial h}{\partial t} \frac{\tan \theta}{\partial \theta / \partial t} \tag{4}$$

Where $h$ and $\frac{\partial h}{\partial t}$ are evaluated on the b-spline curve for each time there is a frequency estimate from spectral analysis. The inital estimates of the parameters $C_1$ and $C_2$ are less important and they are initially set to 0. The parameter $s$ is initially set to 1 mm. The b-spline knot spacing (and hence the number of scaling factors) is constrained by any gaps in the observations: there should not be any gaps larger than the knot spacing as this may lead to instabilities and large errors. A sea level time series at a site with large tidal forcing should theoretically be better captured by a b-spline curve with more frequent knot spacing. However, decreasing the knot spacing also decreases the amount of SNR data that is used to determine the scaling factors.

The reflector height time series that is output from the inverse modelling is converted to a water level time series by taking the additive inverse (reflector height increases as water level decreases), whereby it can be compared with measurements from other water level sensors. The datum could be determined either by installing antennas next to a tide gauge with a visible bench mark or by installing the antennas on a fixed, flat surface and using a levelling device to measure the distance between some mark on the reference antenna and the fixed surface or bench mark.

## 3    Instrumentation

We tested two different types of low-cost GNSS antennas that record data from GPS, GLONASS and GALILEO satellites: TOPGNSS GNSS100L and Beitan BN-84U. These antennas are currently available commercially online for 15–20 USD each. Arrays of four antennas are connected to a Raspberry Pi Zero to log data via USB. To maximise the strength of the multipath interference and to reduce noise from signals received from the coastline, the antennas are attached to a ground plane facing outwards from the coast. Information on how to build and program a similar installation is presented in a separate contribution. The SNR data that is processed for water level measurements is recorded at a frequency of 1 second at a resolution of 1 dB-Hz. A description of the data from the low-cost GNSS antennas and how it is processed for GNSS-R analysis is given in the Supplement. Codes written in MATLAB for processing the raw GNSS data and retrieving water level measurements are provided along with this article.

The key aim of using multiple GNSS antennas is to cancel out noise in the SNR data to improve the precision of water level measurements. It is most convenient and structurally stable to build an antenna array where the antennas are placed side-by-side, but the spacing of the antennas may be important for cancelling out noise. If the source of the noise in the SNR data is due to the local multipath environment then the spacing of antennas should be large enough such that the signal associated with the local environment differs between antennas and cancels out when averaged, with the water level signal remaining. Conversely, if the noise in the SNR data is random instrument noise then the placement of the antennas is likely not important.





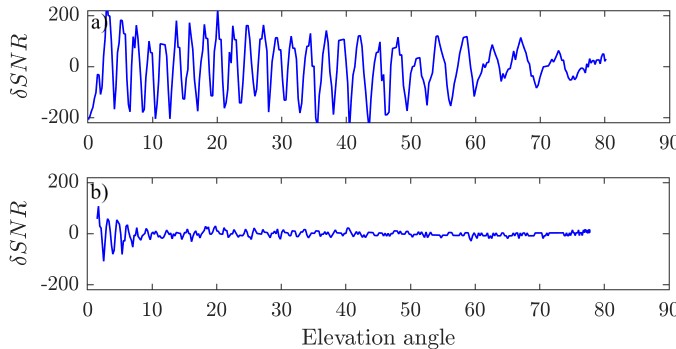

**Figure 1.** Examples of the interference pattern observed in detrended SNR data as a function of satellite elevation angle for a) one of the GNSS100L antennas used in this study placed approximately 3 m above a water surface and b) a Leica AR25 antenna at site GTGU in Onsala, Sweden situated approximately 4 m above a water surface. The data for site GTGU is available online as part of the study Geremia-Nievinski et al. (2019). The oscillations in b) are dampened with increasing elevation angle whereas in a) the oscillations have a constant amplitude.

We therefore tested two different configurations with the antennas spaced apart vertically in a line: two 'tall' configurations where antennas antennas were spaced apart by approximately 25 cm and one 'short' configuration where they were spaced apart as close together as possible (approximately 5 cm). The distance of 25 cm between antennas for the tall configurations was chosen because it is larger than the expected standard deviation of reflector height measurements from each antenna and

therefore the water level measurements from different antennas should occupy a different frequency region at any given time. We also installed both short and tall configurations several meters apart from each other at the same site in order to test if a larger spacing between antennas is important and to test if more than 4 antennas at the same site is advantageous.

As already discussed in Section 1, low-cost antennas that are used in this study are advantageous over geodetic-standard antennas in that they can be used for reflectometry at larger elevation angles. To illustrate this point, we have provided a

comparison of the observed interference pattern using a GNSS100L antenna and a Leica AR25 antenna positioned at a similar height above sea level in Figure 1. Whilst the interference pattern is heavily dampened for more than 10 degrees for the geodeti-standard (Leica AR25) antenna, the oscillations are clear throughout 0 − 80 degrees for the low-cost (GNSS100L) antennas. One disadvantage of the GNSS100L antennas used here and of low-cost antennas in general is that they tend to only record the L1 C/A signal for GPS and GLONASS or the E1 signal for GALILEO and they do not record other signals such as

the modernised L2C and L5 signals that have been shown to be better suited for reflectometry purposes (Tabibi et al., 2015, 2020). Similar GNSS receivers that also utilize the L2C signal are available for approximately six times the cost; these may be investigated in the future.



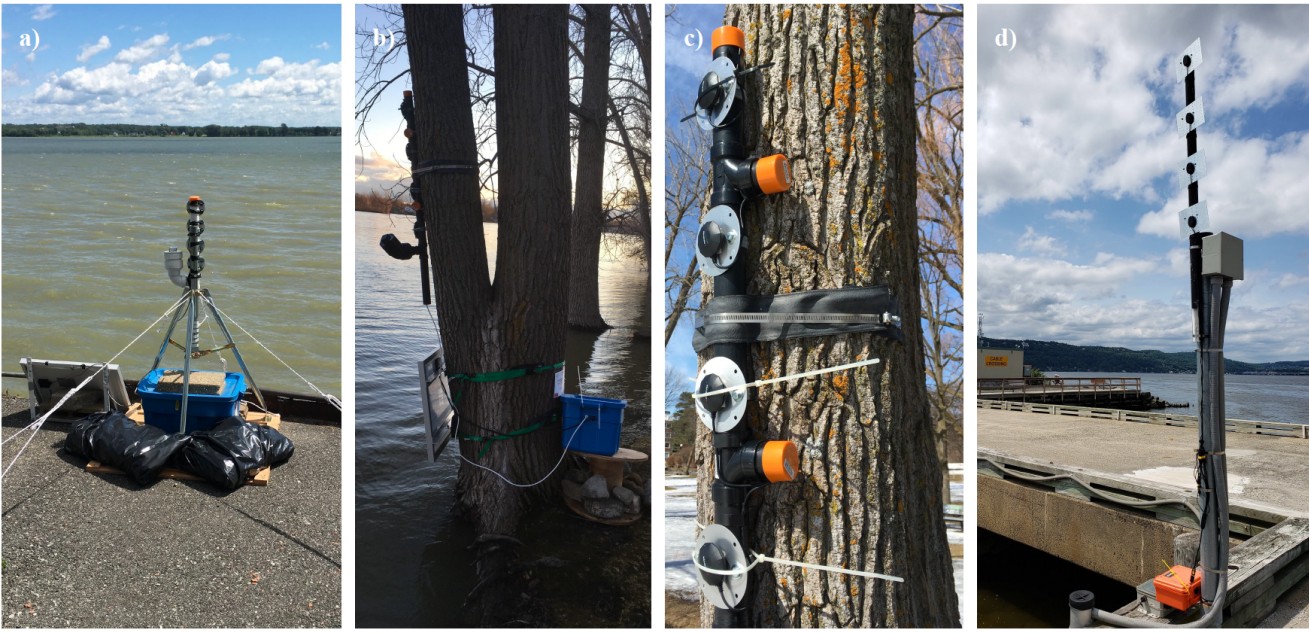

**Figure 2.** Pictures of antenna arrays a) at Trois-Rivières, b) and c) show different angles at Sainte-Anne-de-Bellevue and d) at Piermont. The antenna array showed in b) and c) was also temporarily installed at Trois-Rivières. The antennas are closer together in a) in comparison to the other two setups.

## 4  Test sites

Our experimental antenna arrays were tested at two locations along the Saint Lawrence River in Québec, Canada and one location on the Hudson River in Piermont, NY, USA. Pictures of the three installations are given in Figure 2. There is a water level sensor at each site near to where the antennas were installed. For disambiguation, we will refer to these water level sensors as tide gauges henceforth. The Saint Lawrence River is an ideal testing ground in that water level variations are forced by a range of different signals. From Lake Ontario to the island of Montreal water level variations are dominated by drainage of precipitation, seasonal snow melt and dam activity, whilst tidal forcing becomes progressively more dominant from Montreal to Québec city, where the Estuary to the Atlantic Ocean begins. Daily water level variations at Piermont are also dominated by tides. Key information for the three sites used is summarized in Table 1. The azimuth and elevation angle limits correspond to the region from which the antenna is receiving reflected signals on the water surface and were determined based on visual inspection of oscillations in the SNR data.

The site at which we collected the longest continuous data set is at the Port of Trois-Rivières. This site is on the north shore of the Saint Lawrence river, west (upstream) of the confluence with the Saint Maurice river. Trois-Rivières is approximately midway between Montreal and Quebec City, hence daily water level variations are dominated by tides. There are three OTT Hydromet PLS pressure tide gauges at this site. The average of water level measurements from the three sensors is provided at





**Table 1.** Information about the three test sites.

| Location | Tidal range (m) | Antenna model | Azimuth angle limits | Elevation angle limits | Length of data set | Antenna heights above water level |
|----------|-----------------|---------------|----------------------|------------------------|--------------------|-----------------------------------|
| Trois-Rivières | 0.6 | GNSS100L | 80–220° | 0–50° | 29 days | 4–6 m |
| Sainte-Anne-de-Bellevue | negligible | GNSS100L | 100–240° | 0–80° | 17 days | 2–3 m |
| Piermont | 0.8 | BN-84U | 180–280° | 0–50° | 2 days | 3–5 m |

intervals of 3 minutes from the Canadian Hydrographic Service. According to the instrument handbook, the accuracy is 0.5 cm or less. The short GNSS100L antenna array collected data for a continuous four week period from September 11 to October 9.
Both short and tall GNSS100L antenna arrays were also installed for a 5 day period in August, 2020.

The second site in Québec is in Sainte-Anne-de-Bellevue, at the western tip of the island of Montréal. The stretch of coast where the antennas were situated is part of the northern border of Lake Saint-Louis, which is at the meeting point of the Saint Lawrence and the Ottawa rivers. We installed the antennas on a tree looking over the lake (Figure 2b). The antennas were installed between February and June 2020, however the lake was frozen for the first two months of this period and the antennas
were not continuously recording data due to logistical issues associated with the COVID-19 pandemic. Here we focus on a continuous period of data from May 17th to June 2nd. We used data from a nearby Environment and Climate Change Canada tide gauge to validate our GNSS-R measurements. The tide gauge is a Campbell Scientific CS450 pressure transducer and data is available online at 6 minute intervals from the Government of Canada. It is not clear what specific model of the CS450 sensor is installed at this site, but if we assume that it is a standard accuracy model with 10 m range, the accuracy is at most
1 cm. The tide gauge is situated approximately 500 m west of where the antenna array was installed, just south of a canal and the Sainte Anne rapids that flow from the Lake of Two Mountains to Lake Saint Louis (see Figure 3). We are cautious that differences between the tide gauge and GNSS-R measurements at site could be partly accounted for by differences in the local flow regimes.

At the site on the Hudson River in New York, the antenna array was installed at the end of a large pier. The antennas
were mounted on a pole directly above a Sutron Constant Flow Bubbler for an approximately 48-hour period from 8–10th September, 2020. Data from the bubbler tide gauge at intervals of 15 minutes was downloaded online from the United States Geological Survey. According to the instrument specifications, the accuracy of this sensor is approximately 0.3 cm (0.01 ft).

The RMSE between GNSS-R and tide gauge measurements is used henceforth as a proxy for the precision of the GNSS-R measurements. If we assume that errors from the tide gauge and GNSS-R measurements are not correlated then the RMSE is
actually an upper limit for the precision. It is not a focus of this study to calculate the datum of the GNSS-R measurements relative to that of the tide gauges, hence the mean is removed from each time series and the RMSE is then calculated by evaluating the b-spline curve from inverse modelling at each time that a measurement from the tide gauge exists.



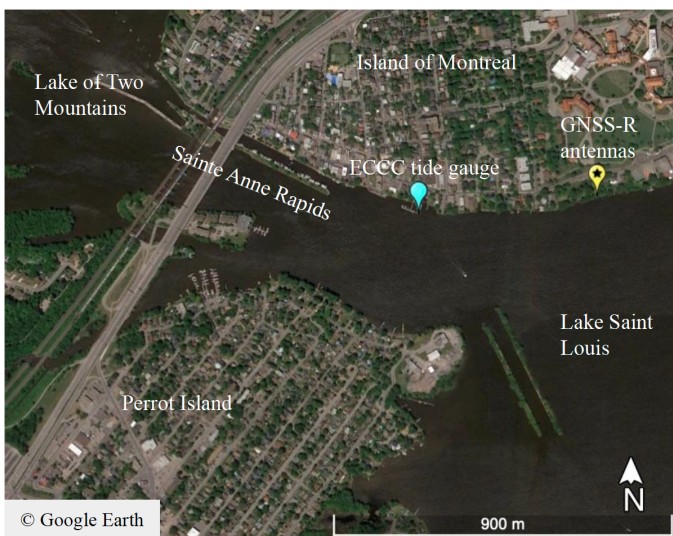

**Figure 3.** The area surrounding the antenna array and tide gauge at Sainte-Anne-de-Bellevue.

## 5 Results

The GNSS-R water level measurements obtained using inverse modelling of SNR data from four low-cost antennas are plotted
alongside tide gauge measurements at all three sites in Figure 4. There is a good agreement between the GNSS-R and tide
gauge measurements at all sites; for the 29-day period at Trois-Rivières the RMSE is 1.02 cm, for the 17-day period at Sainte-
Anne-de-Bellevue the RMSE is 0.69 cm and for the 2-day period at Piermont the RMSE is 1.16 cm. To help visualize the data
that is obtained from the arrays of four antennas, an example of the reflector height solutions from each antenna at Sainte-
Anne-de-Bellevue is given in Figure 5.

The daily water level variations at Trois-Rivières (in Figure 4a) and Piermont (in Figure 4c) are dominated by the principal
lunar semidiurnal tides. However, tidal waves are modified in the shallow river channel at Trois-Rivières such that the crests
travel faster than the troughs, which gives rise to the sawtooth-like pattern at Trois-Rivières as opposed to the smoothly varying
sinusoid that dominates the signal at Piermont. Daily water level variations at Sainte-Anne-de-Bellevue (in Figure 4b) are
of much smaller magnitude as there are negligible tides. The water level variations at this site are driven by drainage of
precipitation and other hydrological effects, for example it rained upstream of Montreal on May 28, which corresponds with a
rise in the observed water level on this day at Sainte-Anne-de-Bellevue.

### 5.1 Influence of multiple co-located antennas

There is an improvement in the precision of water level measurements when using four antennas as opposed to using just one
antenna at all three sites. Reading values from Table 2, if the RMSE obtained with four antennas is compared to the maximum
RMSE obtained with one antenna, there is a reduction of 1.2 cm (or 50%) at Trois-Rivières, 0.3 cm (or 30%) at Sainte-Anne-

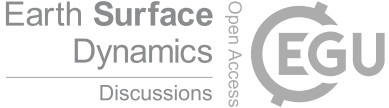

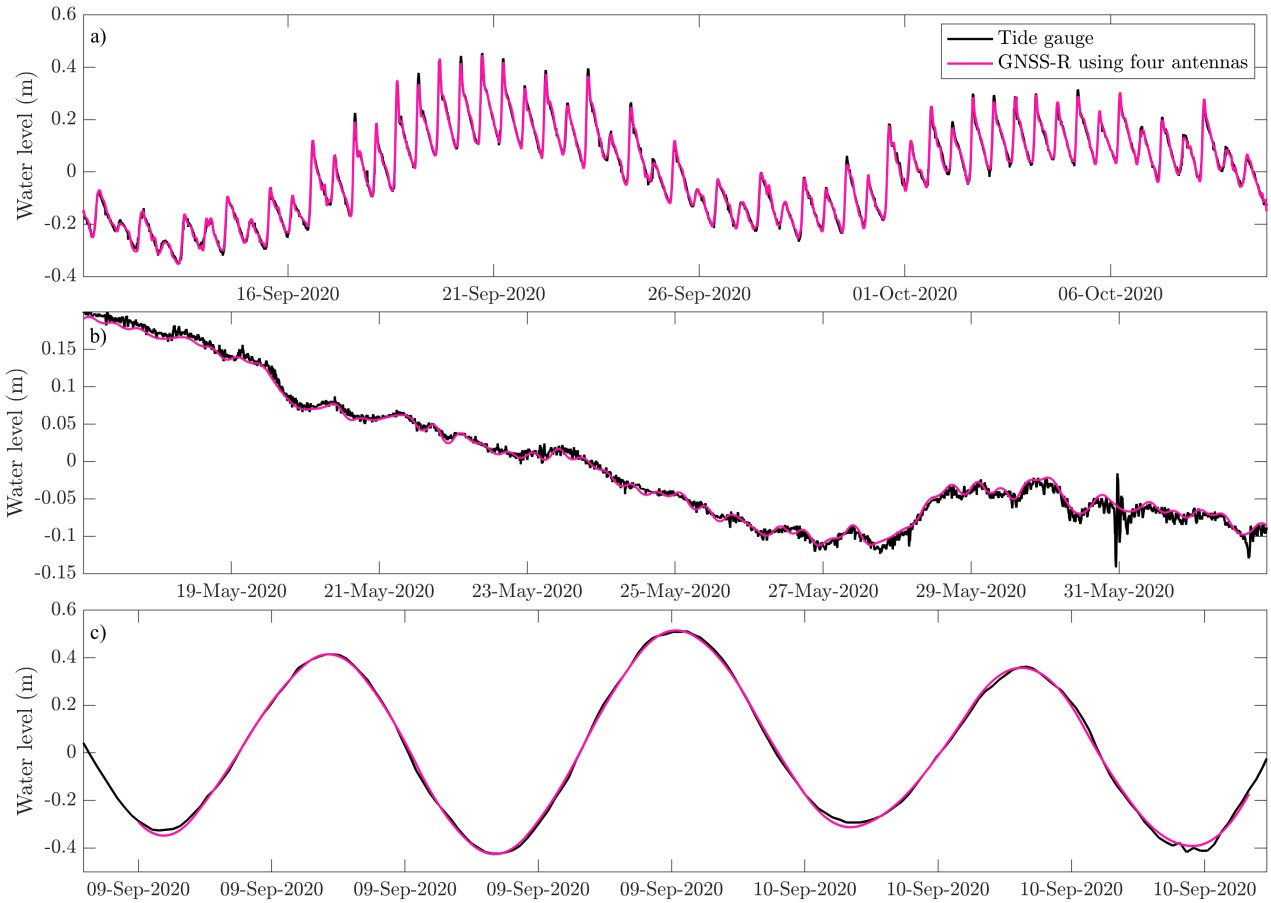

**Figure 4.** A comparison of GNSS-R and tide gauge measurements at a) Trois-Rivières, b) Sainte-Anne-de-Bellevue and c) Piermont. The mean of each time series is removed before plotting.

de-Bellevue and 0.5 cm (or 30%) at Piermont. The results from Table 2 indicate a significant improvement (up to 0.6 cm) when using four instead of two antennas, but minimal improvement between three and four antennas. Four antennas is also recommended over three antennas in case of redundancy. Note that whilst the difference between the minimum RMSE obtained with one antenna is comparable to that obtained with four antennas, our analysis suggests that the relative performance of each antenna appears to be random, which in turn implies that the upper limit should be used for comparison. For example, if the data from Trois-Rivières is split into two two-week periods and analyzed separately, the antennas that give the most and least precise results are not the same for both periods, and neither of these sets in turn match those for the total four week period.

During a period of four days, we performed an experiment with two arrays of four antennas (8 in total) installed at different heights and found that there is no advantage in installing more than four antennas at the same site. For this four day period there is an RMSE of 1.06 cm when using just the short configuration, 1.32 cm when using just the tall configuration or 1.14 cm when using both short and tall simultaneously. Given that the configurations were installed several meters apart, we tested





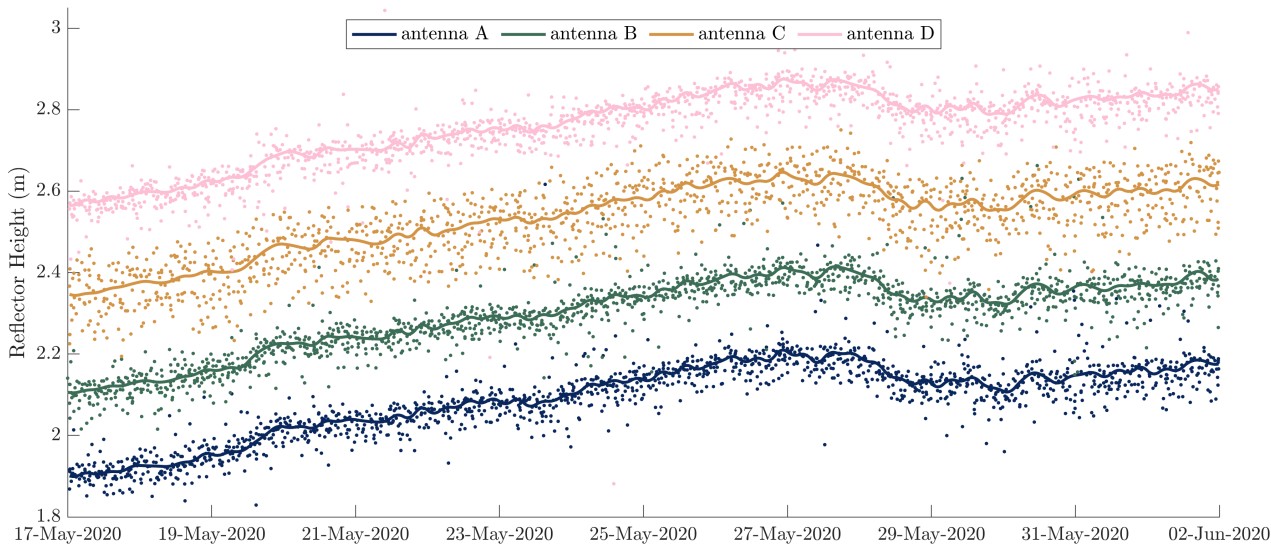

**Figure 5.** Reflector height measurements from four antennas at Sainte-Anne-de-Bellevue. The solid lines represent the b-spline solutions from inverse modelling and the dots are estimates from spectral analysis. The b-spline parameters that are used to plot the solid lines in this figure are averaged to form the single solution in Figure 4b. The curves are inverted because reflector height increases as water level decreases.

**Table 2.** Summary of results obtained using different combinations of antennas to retrieve water level measurements. For 1–3 antennas, the RMSE for every combination of antennas at each site was calculated but only the lowest and highest values are shown in the table.

| | RMSE (cm) obtained using inverse modelling with: | | | |
|---|---|---|---|---|
| Site | 1 antenna | 2 antennas | 3 antennas | 4 antennas |
| Trois-Rivières | 1.24–2.25 | 1.26–1.64 | 1.04–1.10 | 1.02 |
| Sainte-Anne-de-Bellevue | 0.63–0.95 | 0.64–0.81 | 0.65–0.73 | 0.69 |
| Piermont | 1.08–1.67 | 1.07–1.44 | 1.10–1.25 | 1.16 |

different combinations of antennas across both configurations to see if the spacing of antennas influences the precision when using multiple antennas and found that there is no advantage in using antennas spaced further apart than the short configuration. These findings together with our analyses at all three sites also indicate that there is no clear dependence on the heights of the antennas above the water surface. These results also suggest that the source of noise in the SNR data is likely due to random instrument noise as opposed to interference from the local environment.

Our results so far have focused on inverse modelling of SNR data, but spectral analysis estimates, shown as dots in Figure 5, can also be compared with the tide gauge measurements. The RMSE of hourly means of spectral analysis estimates at Sainte-Anne-de-Bellevue varies from 2.4 – 3.9 cm for each antenna. The hourly means can then be combined into a single time series by removing the height offset from each antennna and taking the median of the hourly values, which gives an RMSE



of 1.9 cm. The maximum improvement of 2 cm when using four antennas as opposed to a single antenna for spectral analysis estimates is much larger than the 0.3 cm maximum improvement when using inverse modelling at this site. These differences between inverse modelling and spectral analysis suggest that random noise is a larger source of uncertainty when using spectral analysis, thus supporting the results from Purnell et al. (2020). These results demonstrate that using multiple antennas in the same location is an effective technique for improving the precision in GNSS-R water level measurements, regardless of the technique that is used.

## 6 Discussion of GNSS-R parameters

The results described above were obtained after exploring a large parameter space related to the inverse modelling and experimental setup. The most important parameters to consider are the elevation angle limits, the azimuth angle limits, the b-spline knot spacing and time window length, the satellite constellations used and the temporal resolution of data. We discuss the results of tests with these parameters below. The results discussed in this section were all obtained using the inverse modelling technique given in Section 2.

Note that elevation and azimuth angle limits are constrained by the site surroundings and the discussions in Sections 6.2 and 6.1 refer to investigating optimal limits within these pre-defined constraints. A given azimuth and elevation angle corresponds to a Fresnel zone on the reflecting surface where the power of the signal is concentrated. Software for calculating and visualizing Fresnel zones is given by Roesler and Larson (2018). As the elevation angle decreases, the Fresnel zone increases in size and moves away from the antenna, whereas changing the azimuth angle rotates the Fresnel zone laterally around the antenna. The limits given in Table 1 correspond roughly to the region in which the Fresnel zones are on the water surface and where there are no objects obstructing the view of the water surface (e.g., moored boats).

### 6.1 Elevation angle limits

Using low-cost antennas we retrieved water level measurements over a range of elevation angles up to 50 degrees at Trois-Rivières and Piermont and up to 80 degrees at Sainte-Anne-de-Bellevue, whereas previous ground-based GNSS-R studies have been limited to the use of elevation angles up to 35 degrees. Due to a number of trade-offs between low and high elevation angles, we found optimal limits of 10–50 degrees at all three sites. For sites located at mid to high latitudes (such as our sites), satellites are more prevalent towards the equator (i.e. to the south of our instruments) at lower elevation angles, and therefore using lower elevation angles increases the amount of SNR data that is available for analysis. We find that reflector height measurements are generally less precise at elevation angles greater than 30 degrees, in agreement with previous work using geodetic-standard antennas that found the effect of random noise in SNR data leads to a greater uncertainty at larger elevation angles (Purnell et al., 2020). However, the effect of tropospheric delay increases with decreasing elevation angle and leads to an underestimation of the reflector height (Williams and Nievinski, 2017). For our sites, tropospheric delay has a minor effect on the precision, but afforementioned literature (e.g., Nikolaidou et al., 2019) suggests that this effect is more important at sites where the antennas are at a greater height above the water surface (e.g., > 5 m) or where there is a large tidal range (e.g., > 2 m)





and especially when attempting to find the datum of reflector height measurements (not done here). In general, we find that it is most important to use a large range of elevation angles (e.g., 30 degrees or more) to eliminate any gaps in time in the SNR data

and this is especially important at sites with daily tidal variations, such as at Trois-Rivières and Piermont. We initially tried to use a lower limit of 20 degrees, where the effect of tropospheric delay becomes negligible (Nikolaidou et al., 2019), but found more precise measurements using a lower limit of 10 degrees at Trois-Rivières and Piermont. Limits of 20–50 degrees can be used to make equally precise measurements as 10–50 degrees at Sainte-Anne-de-Bellevue.

## 6.2 Azimuth angle limits

In general, azimuth angle limits should be fixed to match the widest unobstructed view of the water surface. In contrast with elevation angles, there are no complicated azimuth angle-dependent effects; the larger the range of azimuth angle limits, the more data for GNSS-R analysis. The only complicating factor is that some azimuth angles yield more data per day than others because satellites are more prevalent towards the south at mid to high latitudes in the northern hemisphere or vice versa in the southern hemisphere. To demonstrate the importance of maximising the azimuth angle range, we retrieved water level

measurements at Trois-Rivières and Sainte-Anne-de-Bellevue with the azimuth angle range reduce from $140°$ to $100°$. The RMSE increased by approximately 50% (from 1.02 cm to 1.49 cm) at Trois-Rivières and 15% (from 0.69 cm to 0.79 cm) at Sainte-Anne-de-Bellevue when using the reduced azimuth angle ranges.

## 6.3 B-spline knot spacing and time window

For the results described in the previous section, the b-spline knot spacing and time window length vary for each site: at Trois-

Rivières we use a knot spacing of 1 hour and a time window of 6 hours, at Sainte-Anne-de-Bellevue we use a knot spacing of 4 hours and a time window of 24 hours and at Piermont we use a knot spacing of 2 hours and a time window of 6 hours. In general, the knot spacing should be set to 1 hour at a site with large daily tidal variations or a site with a complicated signal (such as the sawtooth-like signal at Trois-Rivières). Note that there is automatically a lower limit on the precision that can be achieved using inverse modelling because a b-spline curve cannot perfectly represent the time series. This lower limit on the

precision increases with increasing knot spacing. For example, if we directly fit a b-spline curve to the tide gauge measurements at Trois-Rivières shown in Figure 4a, we find an RMSE of 0.5 cm if using a knot spacing of 1 hour or 1.6 cm if using a knot spacing of 2 hours. However, decreasing the knot spacing for inverse modelling also decreases the amount of SNR data that is used to compute each b-spline scaling factor, potentially leading to less precise results, hence we do not recommend using a knot spacing of less than 1 hour.

The time window should be at least 3 times the length of the knot spacing. Provided that this minimum is met, the length of the time window does not appear to impact the precision in our experiments. However, both increasing the window length or decreasing the knot spacing greatly increases the computation time for the inverse modelling.



## 6.4 Satellite constellations

Using data simultaneously from all three satellite constellations that are tracked by our low-cost antennas (GPS, GLONASSS
and Galileo) is important to achieve the most precise results. For example, when using only GPS data at Trois-Rivières, the
RMSE increases by over 200% compared to the results obtained using all three constellations (the RMSE is 3.1 cm with just
GPS compared to 1.02 cm with all three constellations). Using any combination of two constellations also leads to less precise
results than using all three constellations: The RMSE increases by approximately 30% (to 1.3 cm) when using just GPS and
GLONASS together or 170% (to 2.7 cm) when using just GLONASS and Galileo together. It is not clear if these differences
are due to the prevalence of satellites (there are more GPS satellites) or differences in the signals themselves.

## 6.5 Temporal resolution

The temporal resolution of SNR data greatly affects the computation time for inverse modelling. Data was recorded at our sites
every second but we found that it was most efficient to resample the data to intervals of 15 seconds prior to analysis. When
varying the temporal resolution between 1 and 15 seconds at Trois-Rivières, we found that the RMSE varies by less than 1 mm
but the computation time increases greatly when using an interval time of one second. The RMSE increases by approximately
70% when decimating the data to intervals of 30 seconds or 60 seconds. Recording data at intervals of 5, 10 or 15 seconds
is therefore advantageous for data storage and efficient analysis. It is important to note, however, that the temporal resolution
provides a limitation on the maximum reflector height that can be resolved due to the Nyquist frequency. See Roesler and
Larson (2018) for a discussion on this Nyquist limit.

# 7 Conclusions

We presented a technique for retrieving precise water level measurements using arrays of four low-cost GNSS antennas and
validated our technique at three sites with variable tidal forcing. By comparing with nearby operational tide gauges, we found
an upper limit on the precision of water level measurements of 0.7–1.2 cm at all sites, whereas previous studies using geodetic-
standard antennas have found an RMSE of 2–50 cm. The values obtained are likely upper limits on the precision because they
also contain error from the tide gauge measurements. These results are significant in that an accuracy of 1 cm is the benchmark
set by GLOSS (2012) for studying multi-year trends in sea level. We found an improvement in the precision of 30–50% when
using four co-located antennas instead of one, which suggests that random noise is one of the key sources of uncertainty in
GNSS-R measurements and supports the results from Purnell et al. (2020). In addition to the reduction in cost (on the order of
several thousand USD or more), we found that the low-cost antennas are better suited for reflectometry than geodetic-standard
antennas because they can be used to obtain water level measurements over a much larger range of elevation angles.

Our results provide a strong proof of concept, but work remains to be done to further validate and improve our technique.
Most importantly, an array of antennas should be installed with a co-located tide gauge for a longer time period (e.g., for at least
several months) and at a site with a larger tidal range. It is also necessary to attempt a levelling procedure so that the datum of



the water level measurements can be defined. To place measurements on a global reference frame and simultaneously monitor
local land deformation, an array of low-cost antennas could then be co-located with a geodetic-standard antenna with the aim
of using the low-cost antennas to obtain precise water level measurements and the geodetic-standard antenna to perform precise
positioning. More models of low-cost antennas should also be tested.

For future installations of low-cost antennas, we propose the following guidelines:

– at least four co-located antennas

– the antennas should record data from GPS, GLONASS and Galileo satellites

– the antennas should be positioned within 1–5 m above the water surface (the spacing apart of the antennas relative to
  each other is not important)

– the antennas should have an unobstructed view of the water surface that extends 140 degrees or more laterally and at
  least 50 metres outwards (this corresponds to the edge of the Fresnel zone for an antenna 5 m above the water surface
  and a satellite at 10 degrees elevation)

– data should be recorded at intervals of 5 seconds (or more, depending on the mean height of the antennas above the water
  surface and the associated Nyquist limit).

Upon obtaining data, the following inverse modelling parameters should be used to extract precise water level measurements:

– a large range of elevation angles (e.g., 10–50 degrees)

– a b-spline knot spacing of 1 hour at a site with tides (or larger would be more efficient at a site with negligible tides)

– a b-spline time window length of at least three times the knot spacing.

The above guidelines may be refined following further field work. For example, we found that four antennas is optimal when
we tested eight co-located antennas for a short period of four days - more data is needed to support this result. Additional data
from other satellite constellations (such as BeiDou) and from other signals (such as L2C and L5) may also be useful.

Our technique for monitoring water levels with arrays of low-cost antennas could be applied to address the need for
widespread, accessible water level measurements in the face of future climate change. The antenna arrays are relatively sim-
ple to build and therefore suited to citizen science efforts. They could be used to obtain sea level measurements in remote
coastal regions as well as lake level or river stage measurements at a fraction of the cost of commercially available sensors but
with comparable precision. Such measurements could be used to validate satellite measurements and to better constrain tidal
models in the polar regions, where there are few coastal sea level measurements. A dense network of sensors could also be in-
stalled to detect spatially variable sea level signals, for example near tidewater glaciers to detect so-called sea level fingerprints
(Mitrovica et al., 2011).

*Code availability.* We are currently preparing to upload codes to Github.





*Author contributions.* DJP, NG and WM conceptualized the study. DJP performed analysis and wrote all codes to go along with the arti-
375 cle. WM designed and built the initial low-cost antenna array (including software). GL, DJP and DP built subsequent antenna arrays and
performed field work. DJP wrote the initial article draft and NG, WM and DP helped to improve the article.

*Competing interests.* The authors declare that they have no conflict of interest.

*Acknowledgements.* The authors would like to thank the Port of Trois-Rivières, the Canadian Hydrographic Service, Fisheries and Oceans
Canada and Environment and Climate Change Canada for help with field work and obtaining tide gauge data.



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
