# Peer review of "Precise water level measurements using low-cost GNSS antenna arrays"

_Earth Surface Dynamics, 2020_

## Referee Comment (RC2)

This work is a valid contribution to the GNSS-R research area. With an antenna array used to calculate water level, authors succeeded in decreasing noise and, consequently, obtaining more precise water level data. This work can be accepted after a minor revision. Below I provide corrections about typos and I also offer some comments on aspects that I believe could enrich the text.

In terms of form, the study's aim, methods, and findings are clear. The title is informative and relevant. The references are relevant, recent, and appropriate studies are included. But in most cases, authors need to review and comment about contribution of each work cited. The state of art is clearly described in the introduction. The research question and scope are clearly outlined. The process of subject selection is clear. The variables are defined and measured appropriately. The study methods are valid and reliable. Tables and figures are relevant and clearly presented. Results are statistically significant. Conclusions answer the aims of the study. The limitations of the study are opportunities to inform future research.
* * *
Major points in the article which need clarification, refinement, reanalysis, rewrites:

1.      About the use of different antenna models or types (TOPGNSS GNSS100L and Beitian BN-84U). Even though both are low-cost, it might be interesting to show their SNR. Do you recommend one model over the other or are both comparable?

2.      You could explain whether the differences found between GNSS-R and tide gauges may be related to the tide gauge technology (pressure or bubbler). There is a difference in the way each of the technologies measure water level. For example, bubblers suffer more influence from the waves.

3.      You could explain if the difference between the tide gauge sampling intervals has an influence on the comparison with the GNSS-R series. Could the increase in RSME (in the order St. A. Bellevue, Trois-Rivières, and Piermont) be related to the fact that the tide gauge sampling interval  increases in the same order (3 min, 6 min, and 15 min)?

4. More details are necessary with regard to the parametrization of reflector height. At least the formula for h(t)=sum(hj*B(t)), with hj scaling factors and B basis should be given (details about the basis can be left out). An interpretation about the scaling factors or coefficients would also benefit the reader – do hj represent the instantaneous reflector height at the knots' time? Finally, in the Methodology authors should inform the reader that a discussion about the postulated time spacing between knots and the time window length is given in the Results section.

5. Section 6.5 is titled Temporal Resolution, but there are actually two types of temporal resolution that could be defined: of the input SNR or of the output water level. The latter is arguably more relevant for the end user and seems to be considered in section 6.3, under an overly technical title: B-spline knot spacing and time window. I suggest complementing those two section titles and reminding the reader that the spline knot spacing can be considered a measure of temporal resolution in the output water level.

6. Authors compare the standard antenna configuration, of one geodetic upright antenna, to an alternative configuration made of multiple low-cost sideways antennas. Therefore, there are three main design factors involved:
A) quantity: one vs. multiple
B) class: geodetic vs. low-cost
C) orientation: upright vs. sideways
While authors focus on factors A and B, factor C should at least be acknowledged, even if it is left for future research. For example, in Fagundes et al. (2021) we have employed a single low-cost zenithal antenna and we are now curious if lateral orientation would have been better.

7. I suspect even the largest antenna spacing considered (25 cm) may be too small to avoid mutual near-field electromagnetic interaction among antennas, depending on the satellite elevation angle considered. This can be understood in terms of overlapping direct first Fresnel zones (dFFZ), ellipsoidal volumes with a focus at the antenna and aligned to the satellite line of sight -- not to be confused with reflection first Fresnel zones (rFFZ), planar ellipses on the surface. In radio wave propagation, dFFZ determines the clearance requirement to minimize interference. While a rigorous treatment may be left for future work, some of the conclusions seem to need caveats. First, the case of an isolated antenna is not really tested, as it is not equivalent to picking an individual antenna in the array, due to the undesirable presence of nearby antennas within the dFFZ. This may partially explain why "the relative performance of each antenna appears to be random". Secondly, the statement that "there is no advantage in using antennas spaced further apart than the short configuration" (5 cm) is only strictly true compared to the tall configuration (25 cm). But the experiments presented do not rule out the possibility that even wider spacings (e.g., 50 cm) could not perform better. Again, the present work is a fine demonstration of the potential of antenna arrays and I recommend it be published, but I am not certain it allows drawing definite conclusions about isolated single antennas and widely spaced antenna arrays.
* * *
Moderate points requiring additional information and suggestions for what could be done to improve the article:

- Authors considered two antenna spacing configurations (25 cm and 5 cm), called "tall" and "short", respectively. I suggest reminding the reader these antenna spacings should not be confused with the higher and lower mean reflector heights considered (5.0 m at Trois-Rivières, 2.5 m at Sainte-Anne-de-Bellevue, and 4.0 m at Piermont).

- In the Introduction, only acoustic and pressure tide gauges are considered. Authors should also discuss radar tide gauges, which represent the state-of-art, and bubbler (pneumatic) gauges, which are used for validation.

- The discussion about normalization of the detrended SNR data would benefit from an illustration of the stronger interference pattern found for GLONASS satellites, at least in the supplementary file.

- How much does the mean separation between successive antennas, computed empirically from reflector heights, differ from the expected separation measured manually with a ruler?

- Authors refer to "two different types of low-cost GNSS antennas", TOPGNSS GNSS100L and Beitian BN-84U. I was curious what receivers were used, but it seems those devices are not just antennas. Rather, each device is a combination of antenna and receiver in the same package, also known as "smart antenna" by GNSS industry vendors.

- Concerning Figure 4, consider showing the differences between GNSS-R and tide gauges in a separate figure, at least as supplementary material. This should provide more insights about the performance, for example: the difficulty in following the sharp sawtooth variations at Trois-Rivières; the outliers near 31-May-2020 at Sainte-Anne-de-Bellevue that may have to be discarded; and an apparent underestimation during low tides at Piermont.

- Consider offering a new table summarizing information about the conventional tide gauges at each site: technology (pressure vs. bubbler), temporal resolution, nominal precision, horizontal distance to GNSS, etc.

- Suggestion for future work: the analysis in section 6.1, Elevation angle limits, could be complemented by considering the spectral analysis retrievals, shown over time in Figure 5, and checking a possible dependence of water level residuals on the satellite elevation angle.

- Consider discussing Yamawaki et al. (2021), "High-rate altimetry in SNR-based GNSS-R: Proof-of-concept of a synthetic vertical array", IEEE GRSL (accepted), DOI:10.36227/techrxiv.14186930

- Consider discussing Williams et al. (2020), "Demonstrating the potential of low-cost GPS units for the remote measurement of tides and water levels using Interferometric

Reflectometry", JOAT, DOI:10.1175/JTECH-D-20-0063.1

- Consider providing a sample of the input SNR as open data as authors state that "A description of the data from the low-cost GNSS antennas and how it is processed for GNSS-R analysis is given in the Supplement. Codes written in MATLAB for processing the raw GNSS data and retrieving water level measurements are provided along with this article."

- RMSE values are given in units of cm, sometimes with one decimal place, other times with two decimal places (e.g., 0.12 cm and 3.4 cm); I suggest adopting units of mm with one decimal place throughout (e.g., 1.2 mm and 34.0 mm).
* * *
Minor points like figures/tables not being mentioned in the text, a missing reference, typos, and other inconsistencies:

In line 38 revise "Tabibi et al. (2020)" to "(Tabibi et al., 2020)"

In the Abstract: Signal-to-Noise ratio -> signal-to-noise ratio [or Signal-to-Noise Ratio]

In the Abstract: "The low-cost antennas are advantageous over geodetic-standard antennas because they are much less expensive ... and they can be used for" -> The low-cost antennas are advantageous over geodetic-standard antennas not only because they are much less expensive ... but also because they can be used for

Line 24: "vary from 0 to 1.7 m" -> may reach up to 1.7 m

Line 38: "the precision … was found to be greater than 1 cm" -> the precision … was found to be lower than 1 cm" [or worse than 1 cm]

Line 41: maybe "interesting".

Line 55: typo in "tropsopheric"

Line 55: "According to Nikolaidou et al. (2019), the tropsopheric delay bias tends to 0 for an antenna  that is 1 – 10 m above a reflecting surface when using elevation angles larger than 20 degrees." -> According to Nikolaidou et al. (2019), the tropsopheric altimetry bias varies from 5 cm to 3 mm for an antenna that is 10 m above a reflecting surface when using elevation angles larger than 20 degrees."

Phrase in line 105: revise the sentence. "it" is a reference to the explanation in lines 102, 103 and 104? "As per Strandberg..."? In this case I think you could insert that explanation together the lines 105, 106 and 107. If it is not true, please reformulate the sentence.

Line 111: about "above", you could specify the equations numbers.

Line 130: 0 and 1 are a default number? Why do you choose those numbers?

Phrase in line 152: Be careful with the punctuation in this phrase. "If the source…".

Figure 1: Unit: Does it convert from dB-Hz to volts/volts and detrended?

Line 136: additive inverse -> negative

Line 144: "the antennas are attached to a ground plane facing outwards from the coast" -> the antennas are attached to a ground plane and oriented sideways, facing outwards from the coast

Line 157: "antennas" are duplicated.

Lines 157 and 158: about "25 cm" and "5 cm", is 25 cm the difference obtained from level average? Or, is it measured manually? If measured, you considered the boards or antenna center?

Lines 167: typo in "geodeti"

Line 160: a different frequency region -> a different multipath frequency band

Table 2: Does the fact that Piermont has a different antenna influence this result?

Line 250: typo in "antennna"

Line 314: typo in "GLONASSS"

Lines 383, 386, 389, 398, 407, 409, 415, 417, 424, 427, 432, 435, 440, 444, 452, 455, 457: References having a DOI should not include URLs to the publisher's website.

Many references have duplicated DOI.

Line 393: additional publication details about Geremia-Nievinski et al. (2019): Journal of Geodesy (2020) 94:70, DOI:10.1007/s00190-020-01387-3

Line 402: Please check this citation.

Line 421: additional publication details about Nikolaidou et al. (2019): Journal of Geodesy (2020) 94:68, DOI:10.1007/s00190-020-01390-8

Line 430: DOI?

Line 445: DOI?

Line 392: additional publication details about Fagundes et al. (2020): GPS Solutions (2021), in press, DOI: 10.1007/s10291-021-01087-1

Line 394: DOI?

---

## Author Comment (AC1)

**Responses to reviewer comment RC1 for esurf-2020-108 "Precise water level measurements using low-cost GNSS antenna arrays"**
By David J. Purnell, Natalya Gomez, William Minarik, David Porter, and Gregory Langston

Below are our responses to the reviewer's comments, with their initial comments in black, our responses in red and quotes from the manuscript indented.

The manuscript is indeed very interesting and presents to my knowledge an innovative approach to GNSS-R . The authors use several low-cost antennas together in kind of an array. They use the well established inverse-modelling technique for their GNSS-R analyses. It appears that using several co-located antennas mounted vertically above each others significantly reduced the SNR noise and thus produces more precise water level results. I think the manuscript is already in a good shape and can be accepted after a minor revision.

We thank the reviewer for their positive feedback and thoughtful suggestions that have helped to improve the article.

I would like the authors to address a few issues in their minor revision:
1. The used term "array" implies a common analysis of the data received with the individual GNSS antennas. However, it seems that the four GNSS-R sensors are analyed completely independently, i.e. not as an array, and then the B-spline coefficients are simply averaged. Please try additionally to analyse the four sensors in one combined inversion directly. Does this improve the performance even more.

The reviewer's understanding is correct that the data from each antenna is initially analysed independently and then the b-spline coefficients are averaged to produce the final time series. We have also tried, as the reviewer suggested, to do a combined inversion using more than one antenna. The results are similar when doing the inversion using more than one antenna compared to doing the inversion with each antenna separately and then averaging the b-spline coefficients. We have added the following text to the supplementary material section S1 where we discuss this combined inversion approach:

> "Instead of using the methodology described in Section 2 in the main text for combining data from co-located antennas by averaging b-spline scaling factors, we also tried performing inverse analysis using data from all co-located antennas simultaneously. This combined inversion is performed by adapting equation (2) in the main text, such that $h$ is written in terms of the reflector height for a chosen reference antenna and the predetermined vertical separation between each antenna and the reference antenna. The RMSE increases from 1.02 to 1.24 cm at Trois Rivières when using the combined inversion approach with all four antennas as opposed to averaging b-spline scaling factors from different antennas. This result suggests that there is no advantage in performing a combined inversion using data from multiple co-located antennas."

And we reference this additional section in the main text in Section 2 of the main text:

> "It is also possible to use SNR data from all four antennas simultaneously as part of the inverse modelling to retrieve a single set of b-spline scaling factors. However, as discussed in the supplement section S1, we found this approach to be less effective."

2. Concerning the temporal resolution, one question is whether the sampling needs to be synchronized, and/or whether there could be advantagous by purposely sampling at different epochs, in partcular when doing a real combined analysis (see question above).

As explained in Section 6.5, we found a limit of 15 seconds for the temporal resolution, passed which (i.e., for greater resolution) results did not significantly improve. We have also found that there is no significant difference between taking a single SNR value every 15 seconds or taking an average of 15 values (one for every second) to produce a single value every 15 seconds. These results suggest that sampling SNR data at different times for different antennas is not likely to improve the results, although it may make the inverse analysis more efficient. The efficiency of our technique is not the primary motivation for this study but should be studied more in the future.

3. Is there any benefit in additionally installing a horizontal array, so kind of a cross installation with both vertical and horizontal extension?

We did not explicitly test a horizontal array, however as explained in Section 5.1, we analysed data from a four-day period where there were two vertical antenna arrays installed several meters apart at Trois-Rivières. Our results suggest that there is no clear advantage in using sets of antennas from both vertical arrays at the same time (i.e., spaced horizontally and vertically).

4. There is at least one typo that I found, line 167 "geodeti-standard", but there might by further typos that I missed. So please check the mansucript carefully with a spell checker.

We thank the reviewer for pointing out the typo and have corrected it. The second reviewer noticed several small typos that we have also corrected.

---

## Author Comment (AC2)

**Responses to reviewer comment RC2 for esurf-2020-108 "Precise water level measurements using low-cost GNSS antenna arrays"**
By David J. Purnell, Natalya Gomez, William Minarik, David Porter, and Gregory Langston

Below are our responses to the reviewer's comments, with their initial comments in black, our responses in red and quotes from the manuscript indented.

This work is a valid contribution to the GNSS-R research area. With an antenna array used to calculate water level, authors succeeded in decreasing noise and, consequently, obtaining more precise water level data. This work can be accepted after a minor revision. Below I provide corrections about typos and I also offer some comments on aspects that I believe could enrich the text.
In terms of form, the study's aim, methods, and findings are clear. The title is informative and relevant. The references are relevant, recent, and appropriate studies are included. But in most cases, authors need to review and comment about contribution of each work cited. The state of art is clearly described in the introduction. The research question and scope are clearly outlined. The process of subject selection is clear. The variables are defined and measured appropriately. The study methods are valid and reliable. Tables and figures are relevant and clearly presented. Results are statistically significant. Conclusions answer the aims of the study. The limitations of the study are opportunities to inform future research.

We thank the reviewer for her positive feedback and detailed comments, which have helped to improve the article.
* * *
Major points in the article which need clarification, refinement, reanalysis, rewrites:
1. About the use of different antenna models or types (TOPGNSS GNSS100L and Beitian BN-84U). Even though both are low-cost, it might be interesting to show their SNR. Do you recommend one model over the other or are both comparable?

We do not recommend either type of antenna yet because we have not had the opportunity to test each antenna side-by-side. The TOPGNSS antennas were tested in Québec Canada and the Beitan antennas were tested in NY, USA by different research groups. Our results suggest that the two different types of antennas perform similarly but we would need to install both types of antennas in the same location to investigate if one type of antenna performs better than the other. We have added the following text to Section 7:

> "More models of low-cost antennas should also be tested, preferably in the same location (we do not recommend one of the types of antennas used in this study over the other type because we have not had the opportunity to test them at the same location)."

2. You could explain whether the differences found between GNSS-R and tide gauges may be related to the tide gauge technology (pressure or bubbler). There is a difference in the way each of the technologies measure water level. For example, bubblers suffer more influence from the waves.

We have added the following text to Section 7:

> " The RMSE values obtained are likely upper limits on the precision because they also contain error from the tide gauge measurements. The amount of error from the tide gauge

measurements is also likely to differ between sites because there are different types of instruments at the sites in Québec (pressure transducers) and Piermont (bubbler gauge)."

3. You could explain if the difference between the tide gauge sampling intervals has an influence on the comparison with the GNSS-R series. Could the increase in RSME (in the order St. A. Bellevue, Trois-Rivières, and Piermont) be related to the fact that the tide gauge sampling interval increases in the same order (3 min, 6 min, and 15 min)?

Our results suggest that the sampling interval of the tide gauge does not have a significant impact on the results. If we increase the sampling interval at Trois Rivières from 3 minutes to 15 minutes (i.e., take every 5th tide gauge measurement and compare with the b-spline output), the RMSE with GNSS-R measurements varies by less than 0.1 mm.

4. More details are necessary with regard to the parametrization of reflector height. At least the formula for h(t)=sum(hj*B(t)), with hj scaling factors and B basis should be given (details about the basis can be left out). An interpretation about the scaling factors or coefficients would also benefit the reader – do hj represent the instantaneous reflector height at the knots' time? Finally, in the Methodology authors should inform the reader that a discussion about the postulated time spacing between knots and the time window length is given in the Results section.

We agree that more information about the b-spline formulation would be useful for the reader. We have expanded our explanation of the b-spline knot spacing in Section 2 as follows:

"...$h$ is represented by a b-spline curve,
\begin{equation}
    h(t) = \sum_{j=1}^{N} h_j B_j (t)
\end{equation}
where $h_j$ are unknown scaling factors, $B_j$ are basis functions and N is dependent on the chosen knot spacing. The scaling factors should be interpreted as control points (as opposed to points along the curve) with a temporal region of influence that depends on the knot spacing (and the b-spline order, which is fixed at 2 here). The b-spline curve is a continuous function that can be evaluated at any time, however the knot spacing controls the amount of scaling factors and hence limits the temporal scale over which features in the water level time series can be resolved. For more information on the b-spline formulation, refer to \citet{strandberg2016})."

We have also added a reference to the results in the Methodology section:

" The influence of the b-spline knot spacing and time window length is investigated in the results Section \ref{bspline_results}."

5. Section 6.5 is titled Temporal Resolution, but there are actually two types of temporal resolution that could be defined: of the input SNR or of the output water level. The latter is arguably more relevant for the end user and seems to be considered in section 6.3, under an overly technical title: B-spline knot spacing and time window. I suggest complementing those two section titles and reminding the reader that the spline knot spacing can be considered a measure of temporal resolution in the output water level.

We have added the description above in our response to point 4 of the b-spline knot spacing for clarity and we have modified the title of Section 6.5 to "Temporal resolution of SNR data".

We agree that the section title "B-spline knot spacing and time window" is technical sounding, however we believe that it may be misleading to name the Section otherwise. The knot spacing is not exactly a control on the temporal resolution.

6. Authors compare the standard antenna configuration, of one geodetic upright antenna, to an alternative configuration made of multiple low-cost sideways antennas. Therefore, there are three main design factors involved:

7.

      6. A) quantity: one vs. multiple
      7. B) class: geodetic vs. low-cost
      8. C) orientation: upright vs. sideways

While authors focus on factors A and B, factor C should at least be acknowledged, even if it is left for future research. For example, in Fagundes et al. (2021) we have employed a single low-cost zenithal antenna and we are now curious if lateral orientation would have been better.

As noted in Section 3, we attach the antennas to a ground plane facing outwards from the coastline in order to reduce unwanted interference from the coast. If the antennas are omnidirectional, then the orientation should not make any difference. We have added the following text to Section 3:

" The antennas used here are assumed to be omnidirectional hence the orientation of the antennas should not matter, but the orientation may be important for other antennas."

8. I suspect even the largest antenna spacing considered (25 cm) may be too small to avoid mutual near-field electromagnetic interaction among antennas, depending on the satellite elevation angle considered. This can be understood in terms of overlapping direct first Fresnel zones (dFFZ), ellipsoidal volumes with a focus at the antenna and aligned to the satellite line of sight -- not to be confused with reflection first Fresnel zones (rFFZ), planar ellipses on the surface. In radio wave propagation, dFFZ determines the clearance requirement to minimize interference. While a rigorous treatment may be left for future work, some of the conclusions seem to need caveats. First, the case of an isolated antenna is not really tested, as it is not equivalent to picking an individual antenna in the array, due to the undesirable presence of nearby antennas within the dFFZ. This may partially explain why "the relative performance of each antenna appears to be random". Secondly, the statement that "there is no advantage in using antennas spaced further apart than the short configuration" (5 cm) is only strictly true compared to the tall configuration (25 cm). But the experiments presented do not rule out the possibility that even wider spacings (e.g., 50 cm) could not perform better. Again, the present work is a fine demonstration of the potential of antenna arrays and I recommend it be published, but I am not certain it allows drawing definite conclusions about isolated single antennas and widely spaced antenna arrays.

Our understanding is that the first Fresnel zone is a concept that is used to determine the radius that needs to be cleared along the path from the satellite to the antenna to avoid

interference. It is not clear how this applies to the situation of co-located antennas possibly interfering with each other. As stated above, the first Fresnel zone is focused at the antenna, hence the radius is 0 at the antenna. First Fresnel zones for co-located antennas will overlap at some point moving outwards from the antennas along the LOS unless they are hundreds of meters apart. A more rigorous investigation of possible interference between antennas would be welcome. We have added the following text to Section 3:

"We note that the distance of 25 cm may not be large enough to avoid interference between antennas."

And in Section 7:

" Whilst our results suggest that the spacing apart of antennas is not important, we cannot rule out the possibility of interference between antennas at the separation distances used in this study. A rigorous investigation of the clearance distance required to ensure that antennas are not interfering should guide a future study."
* * *
Moderate points requiring additional information and suggestions for what could be done to improve the article:
- Authors considered two antenna spacing configurations (25 cm and 5 cm), called "tall" and "short", respectively. I suggest reminding the reader these antenna spacings should not be confused with the higher and lower mean reflector heights considered (5.0 m at Trois-Rivières, 2.5 m at Sainte-Anne-de-Bellevue, and 4.0 m at Piermont).

We have replaced the names 'short' and 'tall' with 'narrow' and 'wide' respectively, throughout the text.

- In the Introduction, only acoustic and pressure tide gauges are considered. Authors should also discuss radar tide gauges, which represent the state-of-art, and bubbler (pneumatic) gauges, which are used for validation.

We have added the following text to the introduction:

" Radar and bubbler gauges are also commonly used to monitor water levels but these instruments are more expensive than pressure transducers or acoustic gauges."

- The discussion about normalization of the detrended SNR data would benefit from an illustration of the stronger interference pattern found for GLONASS satellites, at least in the supplementary file.

Instead of adding a figure, we have added the following text to Section 2:

"This step is taken because the amplitude of the interference in the SNR data varies greatly between different satellite constellations; it is generally stronger for GLONASS satellites. The mean variance of the detrended SNR data for GLONASS satellite arcs is approximately 3 times larger than that of GPS satellites or 6 times larger than that of Galileo satellites."

- How much does the mean separation between successive antennas, computed empirically from reflector heights, differ from the expected separation measured manually with a ruler?

We added the following text to Section 3:

"It is not clear exactly where the antennas are located within the plastic casing, hence these distances were measured from the center of one antenna case to the next (using a ruler). We found that the mean differences between the reflector height time series for each antenna obtained using inverse modelling varied by $\pm$ 1 cm compared to the distances measured using a ruler."

Note that the initial value given for the separation between short antennas (5 cm) was a (bad) guess (we did not have access to the antenna array at the time of writing). We have now measured again and found distances of approximately 10 cm and updated this value in the text.

- Authors refer to "two different types of low-cost GNSS antennas", TOPGNSS GNSS100L and Beitian BN-84U. I was curious what receivers were used, but it seems those devices are not just antennas. Rather, each device is a combination of antenna and receiver in the same package, also known as "smart antenna" by GNSS industry vendors.

- Concerning Figure 4, consider showing the differences between GNSS-R and tide gauges in a separate figure, at least as supplementary material. This should provide more insights about the performance, for example: the difficulty in following the sharp sawtooth variations at Trois-Rivières; the outliers near 31-May-2020 at Sainte-Anne-de-Bellevue that may have to be discarded; and an apparent underestimation during low tides at Piermont.

We have added Figure S1 to the supplementary material showing a shorter period of data from figure 4 with residuals added. We have added the following reference to this figure in the main text, Section 5:

"There is also a shorter period of data for each site plotted with GNSS-R minus tide gauge measurement residuals given in the supplement Figure S1."

- Consider offering a new table summarizing information about the conventional tide gauges at each site: technology (pressure vs. bubbler), temporal resolution, nominal precision, horizontal distance to GNSS, etc.

We do not feel that an additional table is necessary because this information is already clearly stated in Section 4, aside from information about the distance between the tide gauge and antenna arrays at Trois-Rivières, which we have now added:

" The antenna arrays at this site were installed approximately 5 -- 10 meters away from the tide gauges.

- Suggestion for future work: the analysis in section 6.1, Elevation angle limits, could be complemented by considering the spectral analysis retrievals, shown over time in Figure 5, and checking a possible dependence of water level residuals on the satellite elevation angle.

The influence of satellite elevation angles on reflector height estimates due to the influence of tropospheric delay is well-documented and not a focus of this study. We appreciate the suggestion, and will consider a closer examination of reflector height residuals with changing elevation angles in future work.

- Consider discussing Yamawaki et al. (2021), "High-rate altimetry in SNR-based GNSS-R: Proof-of-concept of a synthetic vertical array", IEEE GRSL (accepted), DOI:10.36227/techrxiv.14186930

Thank you for bringing this article to our attention. We have added the reference to the article as follows in Section 7:

> " A vertical array of low-cost antennas such as those used in this study could be used to test the technique for obtaining high-rate sea level measurements recently proposed by \citet{yamawaki2021}."

- Consider discussing Williams et al. (2020), "Demonstrating the potential of low-cost GPS units for the remote measurement of tides and water levels using Interferometric Reflectometry", JOAT, DOI:10.1175/JTECH-D-20-0063.1

We read this article shortly after the initial submission. We have now added the reference in the introduction:

> " Similarly, \citet{williams2020} mounted a low-cost GPS receiver and antenna on the coastline in Ireland near a tide gauge and found an RMSE of 5.7 cm when comparing measurements taken over a 2-year period. The larger RMSE found by \citet{williams2020} compared to \citet{fagundes2020} can be at least partly accounted for by the larger daily water level variations at the coastal site in Ireland due to ocean tides."

- Consider providing a sample of the input SNR as open data as authors state that "A description of the data from the low-cost GNSS antennas and how it is processed for GNSS-R analysis is given in the Supplement. Codes written in MATLAB for processing the raw GNSS data and retrieving water level measurements are provided along with this article."

A sample of one day of data from the 'short' antenna array is given along with codes for MATLAB (and also now translated to python) in the Github link given in the text.

- RMSE values are given in units of cm, sometimes with one decimal place, other times with two decimal places (e.g., 0.12 cm and 3.4 cm); I suggest adopting units of mm with one decimal place throughout (e.g., 1.2 mm and 34.0 mm).

Units of cm are more standard in literature and in instrument documentation. We have updated the values in Section 5 so that they all have two decimal places.
* * *
Minor points like figures/tables not being mentioned in the text, a missing reference, typos, and other inconsistencies:

For the points below we either crossed them out to acknowledge that we made the changes or responded where necessary.

Line 24: "vary from 0 to 1.7 m" -> may reach up to 1.7 m

The lower bound is important because lower bounds have been negative in the past.

~~Line 55: "According to Nikolaidou et al. (2019), the tropsopheric delay bias tends to 0 for an antenna  that is 1 – 10 m above a reflecting surface when using elevation angles larger than 20 degrees." -> According to Nikolaidou et al. (2019), the tropsopheric altimetry bias varies from 5 cm to 3 mm for an antenna that is 10 m above a reflecting surface when using elevation angles larger than 20 degrees."~~

Line 130: 0 and 1 are a default number? Why do you choose those numbers?

We have added the following text to Section 2:
> "The initial estimates of the parameters $C_1$ and $C_2$ are less important and initially set to 0 for computational efficiency. We also found that it is efficient to use 1 mm as an initial guess for parameter $s$."

Figure 1: Unit: Does it convert from dB-Hz to volts/volts and detrended?

We have added units to Figure 1 and also added the following text to Section 2:
> "Observed SNR data (recorded in units of dB-Hz) is converted to detrended SNR data (in units of Watt/Watt) by converting to a linear scale, taking the square root and removing a second order polynomial in $\sin \theta$ space."

Lines 157 and 158: about "25 cm" and "5 cm", is 25 cm the difference obtained from level average? Or, is it measured manually? If measured, you considered the boards or antenna center?

It is not clear exactly where the antennas are located within the plastic casing of the antennas we used, hence we measured from the centre point of the casing from one antenna to the next one. We added the following text to Section 3:
> " These distances were measured from the center of one antenna case to the next (it is not clear exactly where the antenna is located within the plastic casing)."

Table 2: Does the fact that Piermont has a different antenna influence this result?

Answered above in major point 1.

Line 250: typo in "antennna"
Line 314: typo in "GLONASSS"
Lines 383, 386, 389, 398, 407, 409, 415, 417, 424, 427, 432, 435, 440, 444, 452, 455, 457: References having a DOI should not include URLs to the publisher's website.
Many references have duplicated DOI.
Line 393: additional publication details about Geremia-Nievinski et al. (2019): Journal of Geodesy (2020) 94:70, DOI:10.1007/s00190-020-01387-3
Line 402: Please check this citation.
Line 421: additional publication details about Nikolaidou et al. (2019): Journal of Geodesy (2020) 94:68, DOI:10.1007/s00190-020-01390-8
Line 430: DOI?
Line 445: DOI?
Line 392: additional publication details about Fagundes et al. (2020): GPS Solutions (2021), in press, DOI: 10.1007/s10291-021-01087-1
Line 394: DOI?

---

## Referee Report (RR1)

**Review of** "**Precise water levelmeasurements using low-cost GNSS antenna arrays**"
By David J. Purnell, Natalya Gomez, William Minarik, David Porter, and Gregory Langston

Main points:

1. The authors have added the following text to Section 7:

   " The RMSE values obtained are likely upper limits on the precision because they also contain error from the tide gauge measurements. The amount of error from the tide gauge measurements is also likely to differ between sites because there are different types of instruments at the sites in Québec (pressure transducers) and Piermont (bubbler gauge)."

**Reviewer:** Maybe you could be clearer about the magnitude of the errors resulting from the tide gauge. For example, bubblers suffer more influence from waves, so in places where there are higher waves, you have a higher error in tide gauge measurements.

2. As noted in Section 3, we attach the antennas to a ground plane facing outwards from the coastline in order to reduce unwanted interference from the coast. If the antennas are omnidirectional, then the orientation should not make any difference. We have added the following text to Section 3:

   " The antennas used here are assumed to be omnidirectional hence the orientation of the antennas should not matter, but the orientation may be important for other antennas."

**Reviewer:** Commercial GPS/GNSS antennas cannot be assumed omnidirectional. If they were so, one could turn them upside down and they would work equally as well, which is not the case. Actually, they are approximately hemispherical, designed for good reception within +/- 90 degrees from boresight direction. That is the main reason for tipping the antenna sideways in GNSS-R. But there is a tradeoff, as a tipped orientation will have more restricted azimuthal coverage. This may well be left for future work, but it is an important issue.

3. Our understanding is that the first Fresnel zone is a concept that is used to determine theradius that needs to be cleared along the path from the satellite to the antenna to avoid interference. It is not clear how this applies to the situation of co-located antennas possibly interfering with each other. As stated above, the first Fresnel zone is focused at the antenna, hence the radius is 0 at the antenna. First Fresnel zones for co-located antennas will overlap at some point moving outwards from the antennas along the LOS unless they are hundreds of meters apart. A more rigorous investigation of possible interference between antennas wouldbe welcome. We have added the following text to Section 3:
   "We note that the distance of 25 cm may not be large enough to avoid interferencebetween antennas."

And in Section 7:
   " Whilst our results suggest that the spacing apart of antennas is not important, wecannot rule out the possibility of interference between antennas at the separation distances used in this study. A rigorous investigation of the clearance distance required to ensure that antennas are not

interfering should guide a future study."

**Reviewer:** The antenna is at the ellipsoid focus, which lies inside the ellipsoid volume, and should not be confused with the ellipsoid vertex, that lies at the tip of the ellipsoid surface. Thus, the first Fresnel zone surrounds the antenna instead of being entirely in front of it. The extreme case of a satellite at zenith is clearest: for an array, the direct FFZ will be stacked, with the top antennas obstructing the bottom ones. For a satellite at the horizon, the clearance requirement near the antenna would be least. In general, for satellites at an arbitrary elevation angle, the clearance in the direction perpendicular to the line of sight would be converted to the vertical clearance, with the secant of elevation angle. So, higher elevation angles will be compromised more than lower elevations.

[Figure]
* * *
Moderate points:

4. We have added the following text to the introduction:

" Radar and bubbler gauges are also commonly used to monitor water levels but these instruments are more expensive than pressure transducers or acoustic gauges."

**Reviewer:** I think you could write a little bit more about the vantages and advantages of those techniques.

5.   Instead of adding a figure, we have added the following text to Section 2:

"This step is taken because the amplitude of the interference in the SNR data varies greatly between different satellite constellations; it is generally stronger for GLONASS satellites. The mean variance of the detrended SNR data for GLONASS satellite arcs is approximately 3 times larger than that of GPS satellites or 6 times larger than that of Galileo satellites."

**Reviewer:** I believe the figure could be a good contribution to complement the text.

6.   We do not feel that an additional table is necessary because this information is already clearly stated in Section 4, aside from information about the distance between the tide gauge and antenna arrays at Trois-Rivières, which we have now added: " The antenna arrays at this site were installed approximately 5 -- 10 meters awayfrom the tide gauges.

**Reviewer:** I think it could be better to insert a table summarizing the information than the reader searches it in the text.

---

## Author Response (AR2)

Author responses

We thank the reviewer for their additional comments. Below are our responses to the reviewer's comments, with their initial comments in black, our responses in red and quotes from the manuscript indented.

Main points:

1. The authors have added the following text to Section 7:

" The RMSE values obtained are likely upper limits on the precision because they also contain error from the tide gauge measurements. The amount of error from the tide gauge measurements is also likely to differ between sites because there are different types of instruments at the sites in Québec (pressure transducers) and Piermont (bubbler gauge)."

Reviewer: Maybe you could be clearer about the magnitude of the errors resulting from the tide gauge. For example, bubblers suffer more influence from waves, so in places where there are higher waves, you have a higher error in tide gauge measurements.

We have added the following text to section 7:
> "While pressure transducers are more susceptible to errors over long timescales due to instrument drift (Miguez et al., 2005, Pytharouli et al., 2018), bubbler gauges are more susceptible to errors during wavy conditions (Woodworth and Smith, 2003)."

2. As noted in Section 3, we attach the antennas to a ground plane facing outwards from the coastline in order to reduce unwanted interference from the coast. If the antennas are omnidirectional, then the orientation should not make any difference. We have added the following text to Section 3:

" The antennas used here are assumed to be omnidirectional hence the orientation of the antennas should not matter, but the orientation may be important for other antennas."

Reviewer: Commercial GPS/GNSS antennas cannot be assumed omnidirectional. If they were so, one could turn them upside down and they would work equally as well, which is not the case. Actually, they are approximately hemispherical, designed for good reception within +/- 90 degrees from boresight direction. That is the main reason for tipping the antenna sideways in GNSS-R. But there is a tradeoff, as a tipped orientation will have more restricted azimuthal coverage. This may well be left for future work, but it is an important issue.

We removed the quoted text from Section 3. We have added the following text:
> "It should be noted that this configuration would likely reduce the azimuthal range of measurements at a site where there is an azimuthal view of the water surface greater than 180 degrees."

3. Our understanding is that the first Fresnel zone is a concept that is used to determine the radius that needs to be cleared along the path from the satellite to the antenna to avoid interference. It is not clear how this applies to the situation of co-located antennas possibly interfering with each other. As stated above, the first Fresnel zone is focused at the antenna, hence the radius is 0 at the antenna. First Fresnel zones for co-located antennas will overlap

at some point moving outwards from the antennas along the LOS unless they are hundreds of meters apart. A more rigorous investigation of possible interference between antennas would be welcome. We have added the following text to Section 3:
"We note that the distance of 25 cm may not be large enough to avoid interference between antennas."

And in Section 7:
" Whilst our results suggest that the spacing apart of antennas is not important, we cannot rule out the possibility of interference between antennas at the separation distances used in this study. A rigorous investigation of the clearance distance required to ensure that antennas are not interfering should guide a future study."

Reviewer: The antenna is at the ellipsoid focus, which lies inside the ellipsoid volume, and should not be confused with the ellipsoid vertex, that lies at the tip of the ellipsoid surface. Thus, the first Fresnel zone surrounds the antenna instead of being entirely in front of it. The extreme case of a satellite at zenith is clearest: for an array, the direct FFZ will be stacked, with the top antennas obstructing the bottom ones. For a satellite at the horizon, the clearance requirement near the antenna would be least. In general, for satellites at an arbitrary elevation angle, the clearance in the direction perpendicular to the line of sight would be converted to the vertical clearance, with the secant of elevation angle. So, higher elevation angles will be compromised more than lower elevations.

We thank the review for the additional information. We do not feel that this comment requires a change in the manuscript.

Moderate points:

4. We have added the following text to the introduction:
" Radar and bubbler gauges are also commonly used to monitor water levels but these instruments are more expensive than pressure transducers or acoustic gauges."

Reviewer: I think you could write a little bit more about the vantages and advantages of those techniques.

We do not feel that it is important to discuss in detail the advantages and disadvantages of different tide gauge sensors as this is not the focus of this study. We have added the following text to Section 1:
"Radar and bubbler gauges are also commonly used to monitor water levels (see Woodworth and Smith (2003) for a comparison) but these instruments are more expensive than pressure transducers or acoustic gauges."

5. Instead of adding a figure, we have added the following text to Section 2:

"This step is taken because the amplitude of the interference in the SNR data varies greatly between different satellite constellations; it is generally stronger for GLONASS satellites. The mean variance of the detrended SNR data for GLONASS satellite arcs is approximately 3 times larger than that of GPS satellites or 6 times larger than that of Galileo satellites."

Reviewer: I believe the figure could be a good contribution to complement the text.

We considered adding a figure, but (as the reviewer will know), SNR data is very variable for different satellite arcs. We therefore chose to remain with the quantitative information we provided as we feel it is more useful.

6. We do not feel that an additional table is necessary because this information is already clearly stated in Section 4, aside from information about the distance between the tide gauge and antenna arrays at Trois-Rivières, which we have now added: " The antenna arrays at this site were installed approximately 5 -- 10 meters away from the tide gauges.

Reviewer: I think it could be better to insert a table summarizing the information than the reader searches it in the text.

We have added some additional information to Table 1.